# Vitamin D and Obesity/Adiposity—A Brief Overview of Recent Studies

**DOI:** 10.3390/nu14102049

**Published:** 2022-05-13

**Authors:** Imene Bennour, Nicole Haroun, Flavie Sicard, Lourdes Mounien, Jean-François Landrier

**Affiliations:** 1Aix-Marseille Université, C2VN, INRAE, INSERM, 13000 Marseille, France; imene.bennour@univ-amu.fr (I.B.); nicole.haroun@univ-amu.fr (N.H.); flavie.sicard@univ-amu.fr (F.S.); lourdes.mounien@univ-amu.fr (L.M.); 2PhenoMARS Aix-Marseille Technology Platform, CriBiom, 13000 Marseille, France

**Keywords:** preventive nutrition, maternal programming, micronutrients, obesity

## Abstract

Observational studies classically find an inverse relationship between human plasma 25-hydroxyvitamin D concentration and obesity. However, interventional and genetic studies have failed to provide clear conclusions on the causal effect of vitamin D on obesity/adiposity. Likewise, vitamin D supplementation in obese rodents has mostly failed to improve obesity parameters, whereas several lines of evidence in rodents and prospective studies in humans point to a preventive effect of vitamin D supplementation on the onset of obesity. Recent studies investigating the impact of maternal vitamin D deficiency in women and in rodent models on adipose tissue biology programming in offspring further support a preventive metabolically driven effect of vitamin D sufficiency. The aim of this review is to summarize the state of the knowledge on the relationship between vitamin D and obesity/adiposity in humans and in rodents and the impact of maternal vitamin D deficiency on the metabolic trajectory of the offspring.

## 1. Introduction

Vitamin D (VD, here used as a generic term) has a well-established role in the maintenance of phosphate and calcium homeostasis and a clearly essential function in bone and muscle health [1]. For some years now, VD has attracted increasing attention due to the resurgence of VD deficiency in children and adults worldwide [2,3,4] and its suspected multiple extraskeletal benefits, including for cardiometabolic health [1,5]. In the cardiometabolic context, a huge number of in vitro experiments have clearly mapped out the actions of vitamin D on key parameters of adipose tissue and adipocyte biology including adipogenesis and the regulation of gene expression in response to energy homeostasis and inflammation (see [6,7,8] for reviews). These lines of evidence all converge towards a beneficial role of vitamin D in the physiology of adipose tissue, prompting a massive body of research attempting to unravel the relationship between vitamin D and obesity/adiposity in humans and in animals. This review set out to summarize the state of knowledge on this VD–obesity/adiposity relationship and point towards its grey areas and its more robust conclusions.

## 2. Sources and Absorption of Vitamin D

Vitamin D is unique in that it can be both sourced by food and produced endogenously, which means that it has the properties of both a vitamin and a hormone. VD is mainly produced in skin exposed to UVB light, which converts 7-dehydrocholesterol to pre-vitamin D3, which is then further isomerized to vitamin D3 by the action of heat before being released into the circulation [9], where it binds to vitamin D-binding protein (VDBP). The real contribution of endogenous VD production is still under debate: some papers assert that endogenous VD synthesis accounts for up to 70–90% of VD supply whereas others suggest it is just 10–25% of VD supply [10]. The amount of vitamin D3 that is endogenously produced remains highly variable and depends on several factors including latitude, pollution, season, use of sunscreen, clothing, sedentary lifestyle, skin color, and age, among others [11].

Natural exogenous sources of VD are relatively scarce. VD is found in animal-source products (mainly fish liver oil, fatty fish such as salmon, sardines, herring and mackerel, and egg yolk) providing vitamin D3 or ‘cholecalciferol’ [12,13]. Plants and mushrooms are sources of vitamin D2 or ‘ergocalciferol’. Vitamin D (mainly as D3) can also be added in small amounts in fortified milk, margarine or butter, orange juice, and bread or cereals.

After ingestion and emulsion with bile acids [14,15], dietary VD is absorbed in the median part of the small intestine [16]. This process, initially assumed to be passive [17], also requires three apical membrane transporters: Scavenger receptor class B type I (SR-BI), Cluster of differentiation 36 (CD36) and Niemann–Pick C1-like 1 protein (NPC1L1) [18]. A part of the absorbed VD is effluxed by enterocytes via ATP-binding cassette subfamily B member 1 (ABCB1) [19]. The intracellular transport of VD in enterocytes remains unclear, but it ultimately is incorporated in chylomicrons to be secreted in the lymph and delivered to the liver and/or storage organs.

## 3. Metabolism of the Vitamin D

### 3.1. Hydroxylation of Vitamin D and Metabolites

In the liver, endogenous or dietary VD is enzymatically hydroxylated to 25-hydroxyvitamin D (calcifediol, 25(OH)D) (Figure 1). This reaction is catalyzed by microsomal cytochrome P450 enzyme CYP2R1, which is considered the key enzyme of 25 hydroxylation [20,21,22]. However, several other enzymes display a 25-hydroxylation activity on VD, including CYP27A1, CYP2C11, CYP3A4, and CYP2J2 in humans [23,24,25].

Although the 25-hydroxylation is classically thought to be poorly regulated, Bell et al. reported that hepatic 25-hydroxylation was inhibited by 1,25(OH)_2_D and parathyroid hormone (PTH) in humans [26]. Furthermore, several recent studies have described the inhibition of *Cyp2r1* mRNA levels and/or 25-hydroxylation activity in the liver of obese/diabetic mice [27,28,29,30,31] via mechanisms involving several transcription factors including PGC1α and GR [31]. The 25(OH)D thus produced is the major circulating form of VD, with a half-life of about 15 days, and is classically used as a biomarker of VD status [32]. Circulating 25(OH)D is bound to VDBP [33], a serum α2-globulin encoded by the *Gc* gene and synthesized by the liver that is considered as the major plasma protein carrier of VD and its metabolites [34,35]. Indeed, around 80% of plasma 25(OH)D is bound to VDBP, while 19% of 25(OH)D is linked to albumin and the remainder is a free fraction that is thought to be biologically active [36]. The fact that VDBP-null mice remained normal under a VD-sufficient diet suggests that free 25(OH)D levels cover body requirements for physiological functions [37].

The VDBP-bound 25(OH)D is then transported to the kidneys and various other organs and tissues, where it is used for 1α-hydroxylation. In renal proximal tubule cells, the VDBP–25(OH)D complex enters by endocytosis and thus escapes urinary loss [38,39]. This step requires the presence of megalin and cubilin [40]. Cubilin is responsible for sequestration of the VDBP–25(OH)D complex before internalization by megalin. After internalization into vesicles, the VDBP is degraded by lysosomes and the 25(OH)D is handled by intracellular VDBP [41]. The 25(OH)D is then either secreted into circulation or delivered to mitochondria to be metabolized into 1,25(OH)_2_D, the biologically active form of VD. This reaction is catalyzed by 1α-hydroxylase CYP27B1 and stimulated by PTH and low calcium and phosphorus concentrations, inhibited by FGF23, and self-regulated by 1,25(OH)_2_D via a negative feedback mechanism [42].

VD metabolism is ultimately self-regulated via an inactivation pathway that involves CYP24A1-mediated 24-hydroxylation leading to the conversion of 25(OH)D and 1,25(OH)_2_D into 24,25(OH)D and 1,24,25(OH)_3_D, catabolized into inactive calcitroic acid [43]. This inactivation is induced by 1,25(OH)_2_D itself via an induction of CYP24A1.

### 3.2. Mechanisms of Action of Vitamin D

The nuclear receptor-family vitamin D receptor (VDR) is known to mediate the bulk of the biological effects of 1,25(OH)_2_D_3_ [44]. Its ubiquitous distribution explains why a large number of genes (more than 1000) are either directly or indirectly regulated by 1,25(OH)_2_D [44]. VDR heterodimerizes with retinoid X receptor (RXR) and binds to DNA at sites called vitamin D response elements (VDRE) that are located in the promoter regions of VD-regulated genes. In the absence of a ligand, this heterodimer complexes with co-repressors and histone deacetylases, whereas in the presence of 1,25(OH)_2_D, it recruits co-activators and histone acetyltransferases, leading to transcriptional activation [45]. This genomic effect of VD through VDR is strongly suspected to drive a large share of the biological effects of VD in health and disease, notably in the context of obesity, based on evidence from several studies that point to a correlation between VDR polymorphism and pathological issues [46].

The non-genomic effects of VD are characterized by very fast (seconds to minutes) activation of signaling pathways involving phospholipases C and A2, phosphoinositide 3-kinase, protein kinase A, and mitogen-activated protein kinases. It also includes the opening of Ca^2+^ and Cl^−^ channels. These non-genomic effects of VD are dependent on protein disulfide isomerase family A member3 (PDIA3, also known as ERp57, GRP58 and 1,25-MARRS) [47,48], a membrane receptor found in enterocytes, osteoblasts and hepatocytes [49,50,51].

Several epigenetic effects of VD have been described in a number of models and pathophysiological contexts. These effects include DNA methylation, possibly through modulation of DNA methyltransferase and/or DNA demethylase expression [52]. VD can also regulate histone acetylation via activation of histone acetyltransferases and histone deacetylases, but also histone methylation and demethylation, thus modulating chromatin accessibility to transcription factors [53]. Finally, VD is also reported to play a role in regulating the expression of some micro-RNAs (miRNAs) [53], and we recently published data showing that 1,25(OH)_2_D downregulates inflammation-related miRNAs expression in adipocytes both in vitro and in vivo [54].

## 4. Relationship between Obesity and Vitamin D in Rodents

The decrease in plasma 25(OH)D concentrations with obesity is clear in humans, but is less clear in mice (Table 1). Indeed, the decrease in total 25(OH)D is not always obvious, since authors report no modification of 25(OH)D plasma concentration [28,29,55,56,57,58], whereas other studies depict a decrease [27,59,60,61]. The origin of these divergences in mouse studies is still unclear, but could be due to the composition of the high-fat (HF) diet used to induce obesity and/or the methods used to quantify 25(OH)D, i.e., immunoassay tests vs. mass spectrometry. For instance, we initially reported a decrease in 25(OH)D plasma concentrations under HF diet based on ELISA quantification [60], but recent protocols, performed with the same type of diet but using mass spectrometry analysis for 25(OH)D quantification, found no such decrease [29,55].

Other VD metabolites have been measured in obese rodents. We recently reported lower free 25(OH)D rates in obese mice [29,55], which is in agreement with human data [85]. Studies investigating the relationship between obesity and plasma 1,25(OH)2D concentrations in mice have found no clear pattern, as some report decreased plasma 1,25(OH)2D [56,59] whereas others find no change [55] and others even find increased plasma 1,25(OH)2D [28,29]. The pattern probably depends on composition of the HF diet and/or duration of the diet administration.

As is the case in humans, several rodent studies using wild-type or transgenic mice models have attempted to elucidate the impact of diet supplementation with cholecalciferol or active metabolites of VD on obesity. VDR^−/−^ mice were studied to gain insight into the role of VD metabolism on body weight management and adipose tissue biology [62,63,64,65,66]. These mice remain lean and resistant to diet-induced obesity, probably due to the induction of fatty acid oxidation and uncoupling proteins (including UCP1, 2 and 3) in adipose tissue leading to increased energy expenditure. Even if these data strongly suggest that VDR deletion improves energy homeostasis, there are several caveats: the mice were fed a rescue diet containing large amounts of calcium, which is suspected to regulate energy homeostasis [86]; the VDR gene was ablated from the entire mouse, making it impossible to attribute the overall phenotype to a specific tissue [42,68]; VDR^−/−^ mice develop alopecia, and the resulting reduced insulation could increase energy expenditure to maintain body temperature [87].

Other studies have using targeted overexpression or invalidation of VDR in adipose tissue. The overexpression of human VDR in mouse adipose tissue induced an increase in weight and fat pad mass associated with a decrease in energy expenditure and fatty acid oxidation [66,67]. Conversely, adipose tissue-specific invalidation of VDR (Cre recombinase under control of the FABP4 promoter) increased visceral fat pad weight compared to wild-type mice in females only [68] but had no effect on adiposity and body weight in another model (Cre recombinase under control of the adiponectin promoter) [70]. Interestingly, a recent article also reported that adipocyte-specific VDR ablation (Cre recombinase under control of the mouse adiponectin promoter) led to a slight increase in weight gain and an increase in visceral white adipose tissue [69]. Taken together, these observations still give no clear picture of the specific role of VDR in adipose tissue biology and weight management. Furthermore, they also point to a sex-specific effect of VDR that warrants further investigation.

Few studies have been conducted in rodents to evaluate the value or benefit of VD supplementation as a curative strategy. We recently tested VD supplementation in obese mice and found that despite several phenotypical improvements, notably in terms of adipose tissue inflammation, hepatic steatosis and cardiac function, there was no observable improvement in body weight or adiposity [58,74,75], which is in agreement with another study [71]. However, obese mice subjected to 1,25(OH)_2_D injections showed improved body weight and adiposity [72], suggesting that 1,25(OH)_2_D could be more efficient than VD, probably due to the effect of obesity on VD metabolization.

The effect of VD or its metabolites has been more amply described as a preventive strategy against obesity. Several studies have reported a reduction in body weight and/or adiposity in rodents under VD [60,73,74,75] or 1,25(OH)_2_D supplementation [76]. Additionally, it should be noted that VD insufficiency exacerbated adiposity and body weight gain [88,89,90]. The molecular mechanisms are not fully elucidated but may involve an action of VD in the induction of lipid catabolism, notably in the liver and in brown adipose tissue [60]. Interestingly, the ability of 1,25(OH)_2_D to reduce adiposity and induce lipid oxidation in adipose tissue has been confirmed in a *cyp2r1*-deficient zebrafish model [91,92].

Conversely, some studies have reported no effect of VD supplementation or VD deficiency on body weight, adiposity, or adipose tissue fat pad mass [56,57,59,61,77,78,79], whereas other studies reported that VD deficiency decreased HF diet-induced obesity in rodents [80,81,82]. The lack of clear-cut results is problematic, and the origin of these discrepancies remains partly unexplained. Of course, several key parameters are highly variable between studies and could explain the divergent results. Among them are the different quantities of dietary and/or supplemented VD, the quantity and type of lipids incorporated in the HF regimen used, the duration of the regimen and supplementation, and the age of the animals at the beginning of the supplementation are regularly cited, but would require a deeper investigation. Another important issue is the lack of reproducible effect of VD supplementation on plasma 25(OH)D, especially against a background of obesity. Indeed, some authors reported an increase in plasma 25(OH)D [58,59,60,83,84], whereas others reported no effect under VD supplementation [56,57]. It is, therefore, conceivable that the lack of effect of VD on obesity parameters coincides with an absence of detectable increase in 25(OH)D. Note that the difference in concentrations of VD used for supplementation could partly explain these discrepancies, but is clearly not the sole factor. Diet composition and level of 25(OH)D at baseline could also play a major role, as recently highlighted in the case of Mediterranean diet [93], but further research is needed.

## 5. Relationship between VD and Obesity in Human Observational Studies

A large set of cross-sectional studies recently reviewed in deep [94,95] have pointed out the inverse relationship between low serum 25(OH)D and obesity [96] (Table 2). Indeed, it is assumed that plasma 25(OH)D is inversely correlated with most parameters of obesity, such as body mass index (BMI), total fat mass, subcutaneous and visceral adiposity, and waist circumference [97]. These observations have been found in adults but also in children [98] and in aging people [99]. Furthermore, a recent study found that free plasma 25(OH)D and 1,25(OH)_2_D level are lower in obese subjects than normal-weight subjects (85), as also suggested elsewhere [100,101]. This could be due to the reduced release of 1,25(OH)_2_D from subcutaneous adipose tissue under the isoprenaline-mediated lipolysis that occurs during obesity [102].

Many hypotheses have been put forward to explain the low levels of 25(OH)D (total and/or free) associated with obesity. Differences in lifestyle patterns between people with or without obesity cannot be ruled out, including dietary habits, sedentary lifestyle or clothing habits, and there may also be reduced hepatic synthesis of 25(OH)D due to obesity-associated secondary hyperparathyroidism [26], but the key mechanism appears to be VD sequestration. Indeed, the excess adipose tissue observed during obesity could offer an extended storage site for VD and/or 25(OH)D, leading to low plasma 25(OH)D concentrations [103], but this assumption was challenged by Drincic et al. who reported that 25(OH)D was simply diluted in a higher volume in people with obesity, in line with the volumetric dilution hypothesis [104]. Alternative hypotheses have been put forward that suggest modifications in VD metabolism, notably in adipose tissue, where *CYP2J2* mRNA levels were found to be lower in obese women compared to lean women [105]. A recent review argued for a similar set of possible factors affecting vitamin D levels in aging people with obesity (i.e., sequestration or dilution of VD in adipose tissue, increased catabolism of VD in adipose tissue, reduced 25-hydroxylation and reduced sun exposure) [99].

In accordance with the hypothesis of VD sequestration n adipose tissue, a recent meta-analysis highlighted that obesity reduces the effect of VD supplementation in patients with obesity [106]: serum VD concentrations were found to be −38.17 nmol/L lower in obese subjects compared to the normal-weight group, and increasing VD doses did not significantly increase 25(OH)D plasma concentrations, underlining the urgent need to develop strategies for optimal VD supplementation for people with obesity.

Prospective studies suggest that low plasma 25(OH)D levels are associated with a strong prevalence of obesity in children [107], adults [108,109], and elderly women [110], and low VD intake may predict later obesity and metabolic syndrome [111] and even onset of obesity [112].

Some studies have found relationships between VDR polymorphisms and BMI, adiposity markers, or obesity [113,114,115,116] and other research has found a relationship between BMI and polymorphisms of VDBP and CYP27b1 [117]. However, larger studies failed to demonstrate correlations between polymorphisms in genes coding for key drivers of VD metabolism [118,119]. Interestingly, it has been reported that polymorphisms in VDR may influence changes in visceral fat and waist circumference in people supplemented with VD [46]. Mendelian randomization analysis using genes involved in VD metabolism as instrumental variables (VDBP, DHCR7, CYP2R1 and CYP24A1) suggested that low 25(OH)D has little or no impact on BMI [120], that obesity promoted the reduction in plasma 25(OH)D and that a 1 kg increase in body weight leads to a 1.15% decrease in 25(OH)D. Consistently with these findings, a systemic review and meta-analysis of randomized and non-randomized controlled trials showed that weight loss can improve plasma 25(OH)D concentrations [121], and another meta-analysis reported that a weight loss of approximately 10 kg without VD supplementation could increase plasma 25(OH)D concentration by up to 6 nmol/L [122].

## 6. Relationship between VD and Obesity in Human Interventional Studies

Several randomized clinical trials (RCT) have been implemented and recently meta-analyzed in an effort to establish the causal link between low plasma VD levels and obesity (Table 2). Results contrasted between each RCT depending on experimental design (population recruited, 25(OH)D at baseline, duration of VD supplementation, VD dose, etc.). Two meta-analyses failed to show a beneficial effect of VD supplementation on measures of obesity (BMI, fat mass, percentage of fat mass or lean body mass) [123,124] whereas a third meta-analysis pointed to improved BMI and waist circumference following VD supplementation [125]. This lack of clear-cut results is unfortunately common in RCT testing the therapeutic effect of VD, and Dr. BJ Boucher recently highlighted several highly pertinent ways to address this issue by improving RCT design and analysis to obtain definite answers on whether effects of VD translate into real health benefits [126].

**Table 2 nutrients-14-02049-t002:** Relationship between VD and obesity in humans.

	In Humans	References
25(OH)D plasma levels	Reduced in obesity, inversely correlated to markers of obesity and adiposity	[94,95,96,97,98,99]
Free 25(OH)D plasma levels	Reduced in obesity	[85]
1,25(OH)_2_D plasma levels	Reduced in obesity	[100,101]
Impact of obesity on VD supplementation	Obesity reduced the efficacity of VD supplementation	[106]
Low 25(OH)D predictor for obesity onset (Prospective studies)	Yes	[107,108,109,110,111,112]
Effect of polymorphisms in genes coding for proteins involved in VD metabolism (Genetic studies)	Relationship between obesity and SNP in VDR, VDBP, and Cyp27b1 in small cohorts No link between polymorphisms and obesity in larger cohorts	[113,114,115,116,117] [118,119]
Causal effect of low 25(OH)D on obesity (Mendelian randomization)	No	[120]
Causal effect of obesity on low 25(OH)D (Mendelian randomization)	Yes	[120]
Weight lost increase 25(OH)D plasma levels	Yes	[121]
Impact of VD supplementation on obesity (RCT)	Lack of clear-cut results (2 meta-analysis showing no effect, 1 meta-analysis showing improvement of obesity parameters following VD supplementation	[123,124,125]

VD: vitamin D; 25(OH)D: 25-hydrixyvitamin D; 1,25(OH)2D: 1,25-dihydroxyvitamin D; SNP: single nucleotide polymorphisms.

## 7. Maternal Vitamin D Deficiency and Links to Obesity and Adipose Tissue Biology

Investigations are starting to unravel the role of VD during pregnancy [127,128,129,130]. Adequate VD intake during pregnancy is essential for both maternal and fetal health. Indeed, epidemiological studies describe a large range of adverse maternal, fetal and neonatal outcomes associated with VD deficiency [131], such as increased risk for preeclampsia [132], gestational diabetes mellitus [133], higher risk of small-for-gestational-age at term and reduced-term birth weight, and lower head circumference [134,135,136,137].

### 7.1. Lines of Epidemiological and Clinical Evidence for a Link between VD and Obesity in Offspring

Recent studies on large mother–child cohorts have suggested associations between maternal plasma 25(OH)D content and various parameters characterizing obesity in children. In a cohort of 977 pregnant women, low 25(OH)D concentration at 34 weeks of pregnancy was associated with low percent fat mass at birth and high percent f fat mass in children aged 4 and 6 [138]. Similar results were observed in a cohort of 922 mother–child pairs where increased maternal 25(OH)D plasma levels at 15 weeks of pregnancy was associated with decreased percent body fat in offspring [139]. In another cohort including 4903 mothers and their children, severe maternal 25(OH)D deficiency during pregnancy was associated higher percent fat mass and lower percent lean body mass in children at 6 years [140]. In a cohort of 568 mother–child pairs, boys but not girls born to mothers with 25(OH)D deficiency had higher percent fat mass and lower percent fat-free mass at 5 years of age [141]. In a smaller cohort (292 mother–neonate pairs, low maternal 25(OH)D levels were associated with greater superficial and deep abdominal subcutaneous adipose tissue volume in neonates [142]. Other studies in humans have found that maternal VD status is associated with BMI, weight and waist circumference in offspring. Indeed, 25(OH)D concentrations during pregnancy were inversely correlated with BMI and waist circumference in children aged 4 and 6 years [143] but also with high risk for fetal and neonatal overweight [144].

Convergent evidence from all these epidemiological studies therefore tends to confirm the role of maternal VD status in developmental programming of obesity in the offspring. However, long-term studies (follow-up to adulthood) remain scarce; the only study with long-term follow-up (20 years) found no association between maternal VD status and cardiometabolic risk factors [145], and the early mechanisms involved in this programming remain to be elucidated.

RCTs to evaluate the effect of VD supplementation during pregnancy have been conducted and recently meta-analyzed (3725 participants from 22 trials) [146]. Supplementation with VD during pregnancy probably reduced the risks for preeclampsia, gestational diabetes, and babies with low birth weight at term. Another recent meta-analysis (3960 participants from 11 RCTs) found that VD supplementation was also associated with lower BMI and BMI z-score in children aged 3–6 years [147]. However, these data remain weak and the certainty level is only moderate, and so further large RCTs are needed to confirm these results.

### 7.2. Lines of Preclinical Evidence for a Link between VD and Obesity in Offspring

A study conducted in rats showed that maternal VD deficiency induced before and during gestation appears to promote adipocyte and preadipocyte differentiation and proliferation in VD-deficient male offspring [148]. This phenomenon seems to be associated with epigenetic changes (differential methylation of promoters and CpG islets) leading to an obese phenotype (increased body mass and adiposity) in the offspring from VD-deficient females [148]. In mice, maternal VD deficiency is also associated with an obese phenotype in male offspring characterized by higher body mass, adiposity and glucose intolerance but did not translate over to a second generation [149]. Another study also reported glucose intolerance and slower growth in mice born from VD-deficient dam [136].

In a transgenerational study, maternal VD deficiency induced by a VD-deficient diet (from 5 weeks before mating until weaning) lead to disturbances in DNA methylation in somatic liver and germ cells (sperm) over two successive generations. These epigenetic changes were associated with differences in body weight and lean mass-to-fat mass ratio over the two generations [150].

However, this association between maternal VD deficiency and obese phenotype in the offspring is not systematically observed. A study of male Sprague Dawley rats from dams deficient in VD during gestation found no difference in body mass [151], but the offspring showed insulin resistance (high HOMA-IR and lowered glucose tolerance) associated with persistent inflammation (with high plasma and liver levels of IL-1β, IL-6, IL-8 and TNFα). Interestingly, the persistently increased inflammation was explained by the continuously increased IκBα expression related to methylation modifications [151]. Likewise, male mice from VD-deficient dams with intrauterine growth retardation and accelerated growth early in life did not have a higher mass in adulthood [152]. Nevertheless, these animals were predisposed to develop adipocyte hypertrophy in response to a high-fat diet.

We recently reported that juvenile males born to VDD dams had lower body weight and higher energy expenditure compared with controls, whereas females showed no change in body weight [153], which, for the first time, highlights a strong sex-specific metabolic response. Furthermore, we showed that challenging offspring with a HF diet strongly increased adiposity index and insulin resistance in males born to VDD mice, which correlated with insulin resistance, whereas the HF diet-challenged females born to VDD mice had a similar adiposity index and insulin sensitivity to control-diet females. These phenotypes (adiposity and insulin sensitivity) were associated with different transcriptomic profiles in white adipose tissue, prompting us to posit that the specific phenotypic response in females was linked to 17β-estradiol concentrations increased by maternal VD deficiency.

Taken together, these observations support a detrimental role of VD deficiency in terms of mediation of obesity and adiposity that appears to be sex-dependent and also appears to be further exacerbated by HF diet in rodents. However, the molecular mechanisms mediating this phenotype remain elusive.

## 8. Conclusions and Perspectives

The past decade of research has generated very interesting data in transgenic mice and rodents subjected to vitamin D supplementation or restriction, but still without convergent findings. It is vital to pinpoint the origin of these discrepancies, but also to keep in mind that VD deficiency during adulthood is totally different from global embryonic invalidation of the VDR (due to VDR ligand-independent activities and non-genomic effects of VD, among other factors).

Curative approaches based on VD supplementation have recently been implemented in mice but failed to improve obesity or adiposity. Similar results have been found in RCTs on supplementation to patients with obesity, which have mostly failed to confirm a beneficial role of VD supplementation on weight management. Several points for improvement are regularly proposed and should obviously be applied in future RCT designs, in line with Heaney’s guidelines [154]. Trial designs would ideally always include people with measured baseline 25(OH)D and possibly 25(OH)D deficiency in order to highlight benefits of VD supplementation benefit, but these would be ethical issues to address. It would also require supplementing subjects with substantial doses of VD in order to observe a large change in VD status (not just VD intake). Co-nutrient status should also be optimized to limit risks of cofounding factors in the biological response.

Even if the curative role of VD supplementation remains to be established, its preventive role is supported by prospective studies that converge to define low plasma 25(OH)D levels as a predictor of body weight gain. Rodent supplementation studies also partly support the preventive role of VD supplementation in obesity and adiposity. Some studies have nevertheless failed to demonstrate the preventive effect, which makes it vital to understand where such divergence comes from. The inconsistent findings could be linked, at least in part, to an inconsistent ability of VD supplementation to increase plasma 25(OH)D in mice models. Note that this preventive effect of VD is also supported by observational studies describing the impact of vitamin D deficiency in pregnant women on the metabolic programming of their offspring, which are supportive of a preventive metabolic effect of vitamin D sufficiency. Rodent models recently implemented will enable deeper exploration of the metabolic phenotype of offspring from VD-deficient female mice, and raise prospects for unravelling the molecular and epigenetic mechanisms involved in metabolic programming.

To conclude, several lines of evidence are supportive of a preventive effect of VD adequacy on obesity/adiposity, but a potential therapeutic role of VD supplementation for obesity and adiposity remains uncertain. In clinical practice, it is clearly necessary to keep 25(OH)D status within the normal range to avoid potentially associated risks in terms of obesity and adiposity. Well-designed clinical studies and fundamental research are urgently needed to confirm these assumptions.

VD exists in two forms: vitamin D3, which is produced in the skin and found in food, and vitamin D2, which is produced by plants and mushrooms. In the skin, 7-dehydrocholestrol is converted to cholecalciferol following sun exposure. Once synthetized, VD is transported into the circulation bound to VDBP or albumin, whereas dietary VD is absorbed in the median part of the small intestine in a process that requires apical membrane receptors (SR-B1, NPC1-L1, CD36), and is then transported into the circulation incorporated in chylomicrons. For VD to be biologically activated, it needs to undergo two hydroxylations. The first one takes place in the liver, and is catalyzed by enzymes that display 25-hyrdroxylation (CYP27A1, CYP2R1, CYP3A4, CYP2J2) activity. This reaction results in the formation of 25(OH)D which is then transported into the circulation in its free form, and bound to VDBP or to albumin. The second hydroxylation is performed in the kidney and is catalyzed by 1α-hydroxylase CYP27B1, leading to the formation of 1,25(OH)2D. VD metabolism can be regulated: 25-hydroxylation is inhibited by 1,25(OH)2D concentration and PTH, and 1α-hydroxylation is stimulated by PTH but inhibited by FGF23, calcemia, and 1,25(OH)2D via a negative feedback mechanism. Furthermore, VD metabolism is also self-regulated via an inactivation pathway that involves a CYP24A1-mediated 24-hydroxylation, leading to the conversion of 25(OH)D and 1,25(OH)_2_D into 24,25(OH)D and 1,24,25(OH)_3_D which is catabolized into inactive calcitroic acid.

1,25(OH)2D ultimately has many effects, including: (1) genomic effect that requires VDR and RXR and results in regulation of gene expression; (2) non-genomic effects via its association to the receptor 1,25-MARRS and subsequent activation of signaling pathways such as phospholipase C and phospholipase A2, phosphoinositide 3-kinase, protein kinase A, and mitogen-activated protein kinases; (3) epigenetic effects, including miRNA regulation, modulation of DNA methylation, histone acetylation/deacetylation and histone methylation/demethylation.

## Figures and Tables

**Figure 1 nutrients-14-02049-f001:**
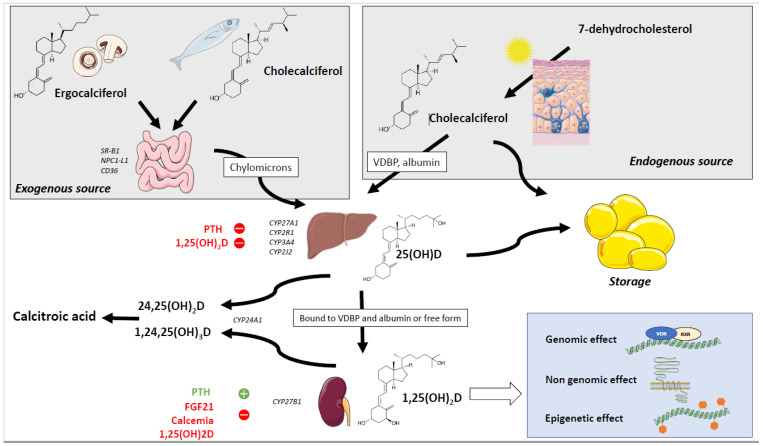
Vitamin D metabolism.

**Table 1 nutrients-14-02049-t001:** Relationship between VD and obesity in rodents.

	In Rodents	References
25(OH)D plasma levels	Lack of clear-cut results (decrease or no effect reported)	[27,28,29,55,56,57,58,59,60,61]
Free 25(OH)D plasma levels	Reduced in obese mice	[29,55]
1,25(OH)2D plasma levels	Lack of clear-cut results (increase, decrease or no effect reported)	[28,29,55,56,59]
VDR^−/−^ mice	Reduced obesity/adiposity	[62,63,64,65,66]
Adipose tissue human VDR overexpression	Increase obesity/adiposity	[66,67]
Adipose tissue VDR^−/−^ mice	Increase obesity/adiposity	[68,69]
	No effect on obesity/adiposity	[70]
Curative effect of VD supplementation on obesity	No	[58,71]
Curative effect of 1,25(OH)_2_D supplementation on obesity	Yes	[72]
Preventive effect of VD supplementation on obesity	Reduction of obesity/adiposity	[60,73,74,75,76]
	No effect on obesity/adiposity	[56,57,59,61,77,78,79]
	Decrease in obesity/adiposity	[80,81,82]
Effect of VD supplementation on 25(OH)D plasma levels in obese rodents	Increase 25(OH) plasma levels	[58,59,60,83,84]
	No effect	[56,57]

VDR: vitamin D receptor; VD: vitamin D; 25(OH)D: 25-hydrixyvitamin D; 1,25(OH)2D: 1,25-dihydroxyvitamin D.

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
