# Peer review of "Vitamin D and Obesity/Adiposity—A Brief Overview of Recent Studies"

_nutrients, 2022, doi:10.3390/nu14102049_

Round 1

Reviewer 1 Report

This is a narrative review on the pathogenetic associations of hypovitaminosis D and obesity. Thw authors have worked experimentally previously on this field. The article is of sufficient quality and design but needs editng with tables and clinical messaged and future research agenda.

I would strongly recommend, adding 2-3 more tables (which I have not found ,although mentioned in the text) highlighting previous animal and human studies,as well as recent results on vitamin D supplementation in obesity.I would also divide the text into a) animal data ,b) human observational and c)human supplementation data as well as the reasons for observed discrepancies.

I would also highlight what's new ,what kind of future designed studies shoul be expected (Heaney's criteria) with a discourse on the daily clinical settin gand future research agenda.

I would also suggest extensive English language editing .

Author Response

Reviewer 1

This is a narrative review on the pathogenetic associations of hypovitaminosis D and obesity. The authors have worked experimentally previously on this field. The article is of sufficient quality and design but needs editing with tables and clinical messaged and future research agenda.

I would strongly recommend, adding 2-3 more tables (which I have not found, although mentioned in the text) highlighting previous animal and human studies, as well as recent results on vitamin D supplementation in obesity. I would also divide the text into a) animal data ,b) human observational and c)human supplementation data as well as the reasons for observed discrepancies.

We are surprised that tables were missing, since we submitted them together with the article. We added them in the main word document in this present version. As recommended we moved the part related to animal studies before human data (observational then interventional).

I would also highlight what's new, what kind of future designed studies should be expected (Heaney's criteria) with a discourse on the daily clinical setting and future research agenda.

Few sentences have been added in the discussion, especially regarding the Heaney criteria.

I would also suggest extensive English language editing.

The English has been corrected by an editing company.

Reviewer 2 Report

Bennour and colleagues have written a nice review regarding the relationship between VD and obesity. I just have some comments and suggestions:

The part of “curative strategy” is well explained and discussed. However, I miss the preventive part. There are no well explained papers talking about the preventive effects of the administration of VD stopping the onset of obesity. Since it is written in the introduction I would add some sentences explaining this part.

From my point of view, this review has too many references (it should have about half approximately) disturbing a bit the flow of the paper. Maybe some of them can be removed, perhaps the redundant ones.

I think the English needs to be corrected. I am not an English native speaker but just to give two examples from the first page in the introduction: the expression “on one hand…on the other hand” hasn’t been used correctly and there is a has instead of a have in the sentence: “… a massive amount of studies has been generated…”.

Author Response

Reviewer 2

Bennour and colleagues have written a nice review regarding the relationship between VD and obesity. I just have some comments and suggestions:

The part of “curative strategy” is well explained and discussed. However, I miss the preventive part. There are no well explained papers talking about the preventive effects of the administration of VD stopping the onset of obesity. Since it is written in the introduction I would add some sentences explaining this part.

This is true that the preventive part is missing especially in human studies ans only prospective studies have been published. We modified the text of the abstract accordingly.

From my point of view, this review has too many references (it should have about half approximately) disturbing a bit the flow of the paper. Maybe some of them can be removed, perhaps the redundant ones.

We agree that there are many references, but it is important to mention all these references since it reflects the quantity of data produced in this field of science and it also shed light on original articles. To our opinion it is particularly fair to mention original articles rather than reviews, even if we could have the feeling that there are too much references.

I think the English needs to be corrected. I am not an English native speaker but just to give two examples from the first page in the introduction: the expression “on one hand…on the other hand” hasn’t been used correctly and there is a has instead of a have in the sentence: “… a massive amount of studies has been generated…”.

The English has been extensively corrected by an editing company.

Round 2

Reviewer 1 Report

 I have no further comments

Author Response

Thank you very much for your second report.

Regards.

JF Landrier